# Nature Forest Reserves in Tanzania and their importance for conservation

Claire Ract[1]*, Neil D. Burgess[1,2], Lars Dinesen[1], Peter Sumbi[3†], Isaac Malugu[4],
Julia Latham[5], Lucy Anderson[6], Roy E. Gereau[7], Marcelo Gonçalves de Lima[8],
Amina Akida[9], Evarist Nashanda[9], Zainabu Shabani[9], Yusuph Tango[9], Someni Mteleka[9],
Dos Santos Silayo[9], Juma Mwangi[9], Gertrude Lyatuu[10], Philip J. Platts[11,12],
Francesco Rovero[13,14]

1 Department of Biology, Centre for Macroecology Evolution and Climate, University of Copenhagen,
Copenhagen, Denmark, 2 United Nations Environment Programme World Conservation Monitoring Centre
(UNEP-WCMC), Cambridge, United Kingdom, 3 TanBE Ltd, Dar es Salaam, Tanzania, 4 Environment and
Forest Certification Limited (EFC), Dar es Salaam, Tanzania, 5 Independent Research Consultant, Exeter,
United Kingdom, 6 Independent Research Consultant, Bath, United Kingdom, 7 Missouri Botanical Garden,
St. Louis, MO, United States of America, 8 Center for Large Landscape Conservation, Cambridge, United
Kingdom, 9 Tanzania Forest Services Agency (TFS), Dar es Salaam, Tanzania, 10 UNDP, Dar es Salaam,
Tanzania, 11 Department of Environment and Geography, University of York, York, United Kingdom,
12 BeZero Carbon Ltd, London, United Kingdom, 13 Department of Biology, University of Florence, Sesto
Fiorentino, Italy, 14 MUSE–Museo delle Scienze, Trento, Italy

† Deceased.
* ractclaire@gmail.com

doi.org/10.1371/journal.pone.0281408

Survey, UNITED STATES

**Data Availability Statement:** All relevant data are
within the paper and its Supporting Information
files.

**Funding:** The authors received no specific funding
for this work.

## Abstract

Since 1997 Tanzania has undertaken a process to identify and declare a network of Nature
Forest Reserves (NFRs) with high biodiversity values, from within its existing portfolio of
national Forest Reserves, with 16 new NFRs declared since 2015. The current network of
22 gazetted NFRs covered 948,871 hectares in 2023. NFRs now cover a range of Tanzanian habitat types, including all main forest types—wet, seasonal, and dry—as well as wetlands and grasslands. NFRs contain at least 178 of Tanzania's 242 endemic vertebrate
species, of which at least 50% are threatened with extinction, and 553 Tanzanian endemic
plant taxa (species, subspecies, and varieties), of which at least 50% are threatened. NFRs
also support 41 single-site endemic vertebrate species and 76 single-site endemic plant
taxa. Time series analysis of management effectiveness tracking tool (METT) data shows
that NFR management effectiveness is increasing, especially where donor funds have been
available. Improved management and investment have resulted in measurable reductions
of some critical threats in NFRs. Still, ongoing challenges remain to fully contain issues of
illegal logging, charcoal production, firewood, pole-cutting, illegal hunting and snaring of
birds and mammals, fire, wildlife trade, and the unpredictable impacts of climate change.
Increased tourism, diversified revenue generation and investment schemes, involving communities in management, and stepping up control measures for remaining threats are all
required to create a network of economically self-sustaining NFRs able to conserve critical
biodiversity values.

**Competing interests:** The authors have declared that no competing interests exist.

## Introduction

Protected areas are essential for conserving biodiversity and maintaining flows of ecosystem services, including storing carbon and providing regular water flows. They are also an important buffer against climate change [1, 2]. The International Union for Conservation of Nature (IUCN) defines a Protected Area (PA) as "An area of land and/or sea especially dedicated to the protection and maintenance of biological diversity, and natural and associated cultural resources, and managed through legal or other effective means" [1].

The creation of protected forest areas in Tanzania has a long history stretching back to the German colonial period in the late 1800s. The 'forest reserves' system expanded over time, first during the German and subsequent British colonial periods and since Tanzanian independence in 1961 [3–6]. Initially, management aims of these areas included reserves established to produce natural forest resources (timber and charcoal), protection of natural forests (water catchment reserves and for the prevention of landslides and erosion), and the establishment of plantation forestry using exotic species.

After the implementation of the 'new' Forest Policy in 1998 [7] and the Forest Act in 2002 [8], national and local authority forest reserves were designed for the preservation of their biodiversity and sustainable use of natural resources and habitats, and human activities were restricted [4, 5]. Under this legislative framework, Nature Forest Reserves (NFRs) were recognised as forest areas of exceptionally high importance for globally unique biodiversity and managed in most cases with solid protection and are recognised as IUCN Category 1b. The first phase of declaring NFRs was in the Eastern Arc Mountains ecoregion [9] during 1997–2009, starting with Amani Nature Reserve in the East Usambara Mountains in 1997.

Over the past 25 years, the Tanzanian government has continued identifying and upgrading other biologically important reserves to become NFRs. These sites initially fell under the ownership and management of the Forestry and Beekeeping Division of the Ministry of Natural Resources and Tourism and (since 2011) the Tanzania Forest Services (TFS) Agency. After the first phase of declaring NFRs, the network expanded to cover all the different forest types in the country (including the Coastal forests, the Northern Volcanics, the Southern Highlands, Guineo-Congolian lowland forests, Miombo woodland and Miombo-Acacia woodland). The current network of NFRs in Tanzania contains 22 reserves with 4 additional reserves being proposed in 2022 (Fig 1). However, our analyses contain only 21 reserves declared, as the raw data is not available yet for (i) the proposed sites (except for Nou NFR) and (ii) one recently declared reserve ((Uvinza), gazetted beginning of the year 2022). Uplifting the existing reserves to the status of NFR was undertaken to facilitate the development of income-generating activities like tourism, to address threats facing some of the reserves that were being managed as part of the network of reserves at regional and district levels, and to recognise the exceptional biological values of sites outside the network of national parks and other reserves managed primarily for large mammal conservation in Tanzania.

We describe and assess the conservation value of Tanzania's NFR network for the first time by (i) evaluating the development of the NFR network and assessing its coverage of Tanzanian biodiversity, focusing on endemic species; (ii) assessing the coverage by the NFR network and other protected areas across Tanzania of endemic and rare species (iii) assessing the effectiveness of NFRs and whether management resources have been deployed to the most important sites to achieve conservation goals. We provide current insights into the management challenges in the NFR and discuss how these might be addressed in the future.

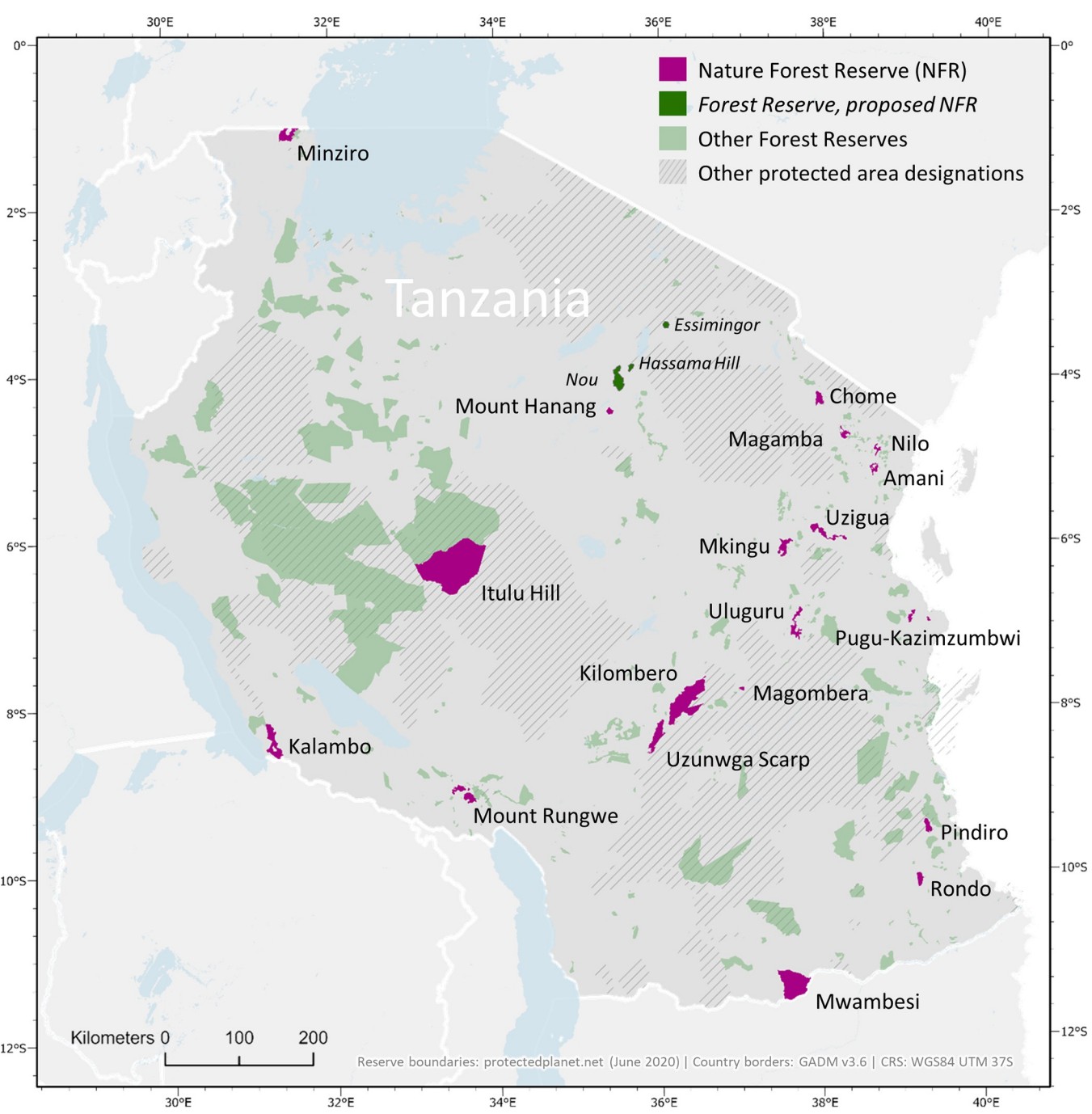

**Fig 1. Map of Tanzania showing Nature Forest Reserves (pink), proposed Forest Reserves (dark green), other Forest Reserves (green) and other kinds of protected areas (shaded).**

## Methods

### Data collection

We collected available published and unpublished data on the Tanzanian NFRs (summarised in S1 Table). Data NFRs include declaration details, biological features, management information, and threats. Management data are available for 18 sites as the five were too recently declared (Pugu-Kazimzumbwi, Uzigua, Essimingor, Hassama Hill and Nou).

## Site level species data

Lists of vertebrate species and plant taxa in each NFR were built upon existing lists of vertebrates for the Eastern Arc Mountains [6, 10–12] and coastal forests [13] and the addition of data from other sites by the authors. Lists of plant taxa in each FNR were created using data from the Missouri Botanical Garden, augmented with records from the GBIF portal (accessed in July 2020) and the IUCN red list. The designation "taxa" is used throughout the text to mean "terminal taxa" for plants, including species, subspecies, and varieties. The unit of analysis for vertebrates is species. Taxa or species occurring in each of the NFRs were categorised as endemic (both endemic to Tanzania or to the NFR itself) and threatened with extinction according to IUCN Red List categories: Critically Endangered (CR), Endangered (EN), and Vulnerable (VU).

## Tanzania wide species data

Vertebrate species distribution data for the whole of Tanzania were compiled from existing GIS data of potential distributions: birds [14], amphibians and mammals [15], and reptiles [16]. Species occurrences within NFR were calculated using GIS analysis of species range distributions overlaid on NFR shapefiles from the Word Database on Protected Areas (https://www.protectedplanet.net/en).

## Management data

Declaration date and area for each NFR were gathered from Tanzanian government records. Assessments of NFR management effectiveness were done using the Management Effectiveness Tracking Tool (METT) [17–19]. METT assessments were applied four times for 11 NFR; in 2015, 2017, and 2019—and for 17 NFR in 2021. To assess management effectiveness further, we used data on forest disturbance collected for four NFRs (Amani, Kilombero, Mkingu, and Uluguru) where baseline (2001—pre NFR creation) and more recent (2019) survey information exists. Disturbance data were gathered using 10 m wide by 600 m long transects within the forest (a total length surveyed of 3.3km; [20]). Data on living, naturally dead, or cut stems of trees were collected. Repeating disturbance transects before and after reserve declaration aimed to determine whether the declaration of NFR status is correlated with reduced habitat degradation.

Data on income generation and tourism trends were collated from the managers of each NFR, reserve management plans, and from unpublished baseline, mid-term, and end-line NFR surveys undertaken as part of a Global Environment Facility (GEF) project "*Enhancing the Forest Nature Reserves Network for Biodiversity Conservation in Tanzania'*. In 2019, compiled data were checked by managers of all 17 NFRs and the staff of UNDP-Tanzania and the Tanzania Forest Service [21]. Income was standardised to US dollars across the years assessed. Some further updates on tourism and funding were provided by Tanzania Forest Service staff in June 2023 based on July 2022-June 2023 data.

## Data analysis

We assessed the biodiversity value of individual NFRs as the number of both endemic and threatened taxa in each reserve using our database of species-per-reserve. Arranging reserves in chronological order of declaration facilitated an assessment of how biodiversity value has been added through the declaration of new NFRs. The correlation between the number of endemic and threatened taxa per NFR and the declaration date was tested using the non-parametric Spearman's rank correlation coefficient.

We assessed Tanzania endemic species coverage by the NFR network using range data from the IUCN Red List for amphibians, reptiles, birds, and mammals overlaid onto GIS data for NFRs. NFRs were grouped according to the date they were established, resulting in eight groups/time frames (as several reserves were declared the same year). For each endemic species, the percent of its range covered by the sites was determined and classified into different categories: (1) gap species: 0% of an endemic species' range covered by the sites, (2) poorly covered species: between 0–2% of an endemic species' range covered by the sites, (3) 2–5% of an endemic species' range covered by the sites, (4) 6–10% of an endemic species' range covered by the sites, (5) between 11–20%, (6) between 21–50%, and (7) more than 50% of an endemic species' range covered by the sites. The proportion of endemic species was calculated in each of these categories. We also mapped the number of endemic species not included in the NFR network as it was developed over time.

In addition, we used reserve polygons from the World Database on Protected Areas (accessed in April 2022) to determine if there are other types of protected and conserved areas (National Parks, Game Reserves, Forest Reserves, Conservation Areas, Village Land Forest Reserves, and Game Controlled Areas) that overlap with the ranges of gap/poorly covered species once species covered by the 23 NFRs are excluded. The percent species range covered by different protected area categories was calculated, as was the proportion of species in each of the classification categories. Analyses were carried out using the statistical software R (version 4.0.5) [22], the spatial Geographic Information System QGIS (version 3.16.14), and Microsoft Excel.

For management effectiveness and other aspects of reserve management, possible correlations among the covariates were tested using Spearman rank correlation, Friedman test and the post hoc pairwise Wilcoxon rank sum test in R [22].

## Methodological limitations

The data and analyses used here are subject to several limitations.

Firstly, for the management effectiveness analyses there is limited independent verification of the METT data from reserve managers, and past METT data on recently declared NFRs were missing. Earlier studies have shown that METT data are valuable but need to be interpreted with care when done using self-assessment [17, 23], especially older METT versions, i.e., METT 1–3. To obtain the best outcome from the METT assessment process, studies suggest that METT assessment is done by a group of people to reduce bias [19, 24]. In addition, METT 4 requires that local communities are involved in the assessment. Our METT data used different versions of the tool which we combined. Management effectiveness assessment benefited from field surveys of forest disturbance, but these were only available for four NFRs.

Secondly, for the species gap analyses, we used range data only, without considering life history or population data. This means that the viability of populations within a NFR or more broadly is not examined. Species range data may also be biased, especially for areas that have been poorly sampled [25].

Thirdly, undertaking analyses using current species occurrence data does not consider possible range changes due to climate change. A study based on modelling the impact of climate change on Tanzanian forests demonstrated that most forest ecosystems, especially montane forests in the Eastern Arc Mountains will be affected with habitat losses of more than 40% under the optimistic RCP4.5 scenario by 2055 [26].

## Results

### Biodiversity values of NFRs using site level data

Analysis of our species-by-sites database shows that older NFR sites supported more threatened taxa than recently declared sites (Fig 2A and S2 Table). Amani, Kilombero, and Uluguru

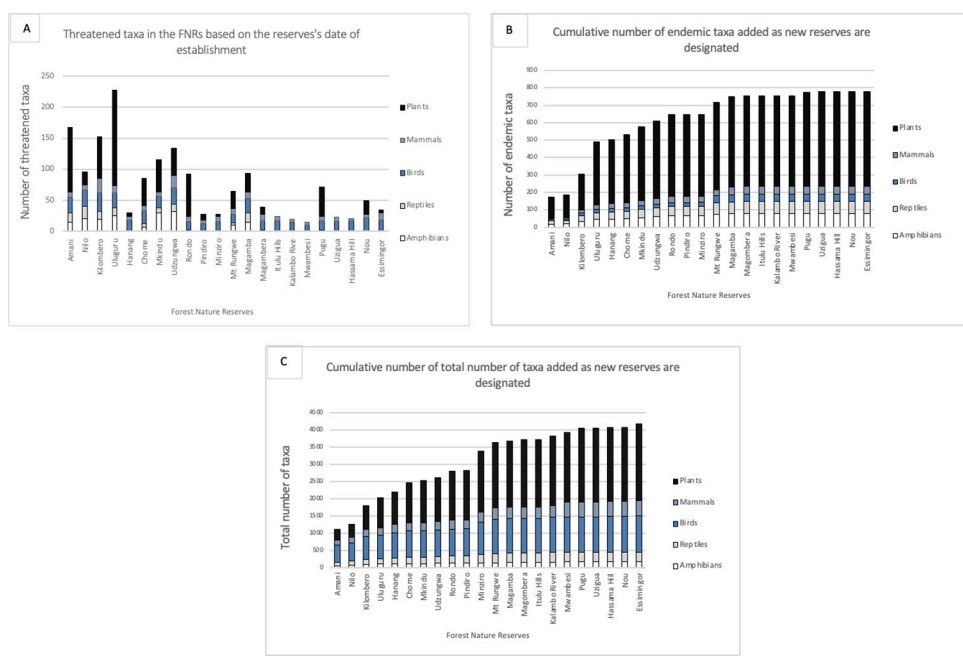

**Fig 2. Importance of the network of NFRs for Tanzanian endemics and globally threatened taxa since 1977.** A: Number of threatened taxa in the NFRs. B: Cumulative number of endemic taxa. C: Cumulative number of total taxa (including of endemic, threatened, and other taxa). Reserves are ordered according to the year of their designation. The stacked bars are separated depending on the taxa: black represent plants, light blue is for the mammals, dark blue represent the birds, grey is the reptiles and white symbolize the amphibians.

NFRs had the highest number of threatened taxa, followed by Nilo, Chome, Mkingu, Uzungwa Scarp, Rondo, and Magamba. Many of the threatened species were also endemic to Tanzania (see Fig 2B, Amani to Mangula). As expected, as new reserves were added to the network there was a gradual increase in the number taxa in the NFR network, but this was found to asymptote for all reserves declared since 2019 (Fig 2C). The cumulative number of endemic taxa was correlated with total taxa and date of declaration of the reserves (Fig 2, $r_s$ = 0.97, p < 0.001 for all taxon groups). This significant increase is also found in each taxon group separately ($r_s$ = 0.95, p < 0.001 for each taxon group).

## Coverage by NFRs of Tanzanian endemic and threatened species using range map data

Our GIS analysis using IUCN species range data, shows that as the number of NFRs increased, the number of Tanzania endemic vertebrate species covered by reserves also increased, and the number of gap species declined. For the 26 Tanzanian endemic bird species, all species now have part of their range covered by NFRs (Fig 3A). Of the 26 endemic mammals, the number of endemic gap species declined to 5% when all sites were included (Fig 3B). More than 25% of 65 Tanzanian endemic amphibian species ranges are partially covered by the NFRs (corresponding to more than 50% of the species range covered) (Fig 3C). However, amphibians contained the highest proportion of gap species for all sites combined (Figs 3C and 4C). The proportion of 63 endemic reptile gap species declined to 10% when all NFR were added (Fig 3D).

The richness of endemic gap species declined as more NFRs were added to the network. However, the number of poorly covered endemic species increases, as more species are

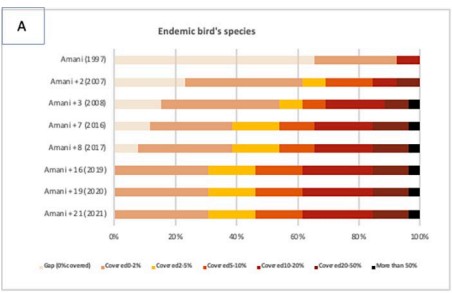
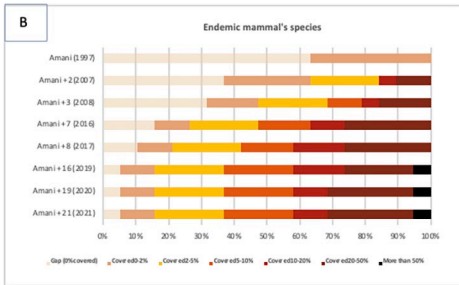
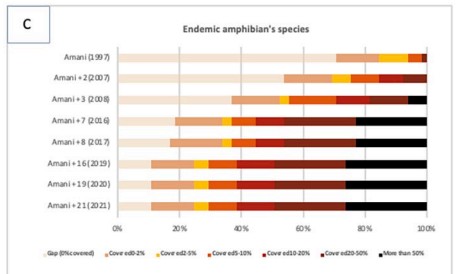
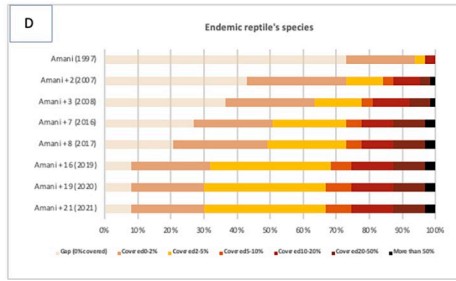

**Fig 3. Proportion of Tanzanian endemic vertebrate species ranges covered by NFRs over time.** Years when the reserves were gazetted are in parentheses, and the number of reserves declared in each time frame is indicated after "Amani". The light bars refer to the endemic gap species, where 0 per cent of their range is covered. Other colours show the percentage of the endemic species range covered by the reserves. It goes form light brown, between 0 and 2 percent is covered, yellow between 2 to 5 percent covered, orange between 5 and 10 percent, dark red, between 10 and 20 percent, dark brown, between 20 and 50 percent, and black, where more than 50 percent of the species range is covered by the reserves.

included with only a small part of the range covered by NFRs (Fig 4D–4F). The most notable remaining coverage gaps after all 23 NFRs are included are for endemic reptile species in south-eastern Tanzania (Rondo and Pindiro areas) and central areas of the country. The most notable remaining gap for amphibians was a species of frog (*Hyperolius puncticulatus*), endemic to the island of Unguja (often known as Zanzibar) (Fig 4C).

If all types of protected areas present in Tanzania are included in the analysis, the proportion of gap and poorly covered endemic species declines further (Fig 5). For birds, there are no poorly covered species across all protected areas, compared to 30% poorly covered bird species in NFRs (Table 1). However, for amphibians, almost 7% of endemic amphibian species remain unprotected even when all protected areas are considered, but this is reduced from 11% for just NFRs (Table 1).

### Effectiveness of NFR management in achieving conservation goals

The effectiveness of management of the NFR network, as assessed using METT data, increased over time. For the 11 NFRs present in 2015 the METT percent was 55%, 69% in 2017 for 12 NFRs, 69% in 2019 for 17 NFRs assessed, and in 2021 increased to a mean score of 87% for the 17 NFRs evaluated (Fig 6A). Management effectiveness has improved in recent years, and each year was statistically significant different from the other (Friedman's test, chi-squared value = 37.789, df = 3 and p value = 3.133e-08) (Fig 6A). Furthermore, the statistically significant difference of METT scores was the greatest between the year 2015 and 2021, following our initial hypothesis that the average management effectiveness has improved over the years (Pairwise Wilcoxon rank sum test, p value between the year 2015 and 2021 = 2.6e-05). Total tourist numbers (mainly international visitors) increased steadily in most NFRs, from 1,698 tourists in 2016 to 4,097 tourists in 2019 (Fig 6C; rs = 0.95, all p values < 0.0123, N = 17). New data post the COVID-19 Pandemic shows that in 2022 a total of 242,824 tourists visited NFRs

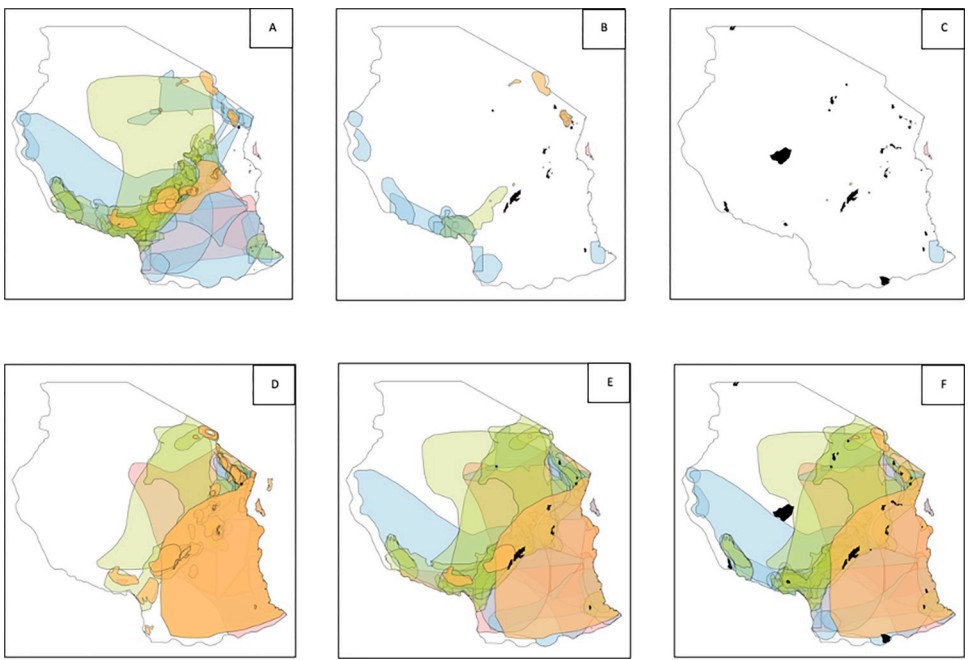

**Fig 4. Coverage of the range of Tanzanian endemic 'gap and poorly covered' endemic species in 1997, 2017 and 2022.** Coverage of the range of Tanzanian endemic 'gap' species (panels A, B, C) and poorly covered endemic species (panels D, E, F). Panels A and D show endemic species coverage in 1997 when the first reserve was added to the network. Panels B and E show the endemic species coverage in 2017, with 9 reserves in the network. Panels C and F represent the species coverage in 2022, with 19 NFRs and 3 proposed NFRs included in the network. The orange coverage range displays endemic mammal species, blue reptile species, green bird species, and light pink amphibian species. The NFRs are indicated as black polygons.

and brought in a total of TZS 1.360,940,965 (about 550,000 USD). Income generated decreased slightly from 2017 to 2019, and again in 2021, due the effects of COVID-19 Pandemic (Friedman's test, chi-squared value = 10.478, df = 3 and p value = 0.01491) (Fig 6D). However, the results from the post hoc tests showed no statistical difference of the income generated between the four years (it should be noted that our dataset contained a lot of unavailable data, especially for the more recent sites).

Management capacity, including the number of buildings, transport, and equipment present in each NFR increased between 2015 and 2019, but the number of staff and rangers declined in 2019 (Fig 6E). Only office equipment (corresponding to the number of computers, photocopier scanners, printers, GPS units, solar batteries, and hard drives) increased between 2015 and 2019.

Analysis of forest disturbance transect data from four NFRs in 2001 (before these NFRs were declared—except for Amani) and 2019 shows that numbers of cut trees and cut poles per hectare have declined since the reserves were declared (except for Uluguru and Kilombero NFRs) (Fig 7). The number of live trees was stable (in Uluguru NFR) or decreased (in Kilombero and Mkingu NFRs). In contrast, the number of live poles increased (except for Kilombero NFR). However, none of the results were significant.

## Discussion

### How well do NFRs capture Tanzanian biodiversity values?

The existing system of NFRs was not formally designed using strategic conservation planning approaches, but was based on a combination of expert knowledge, the framework of

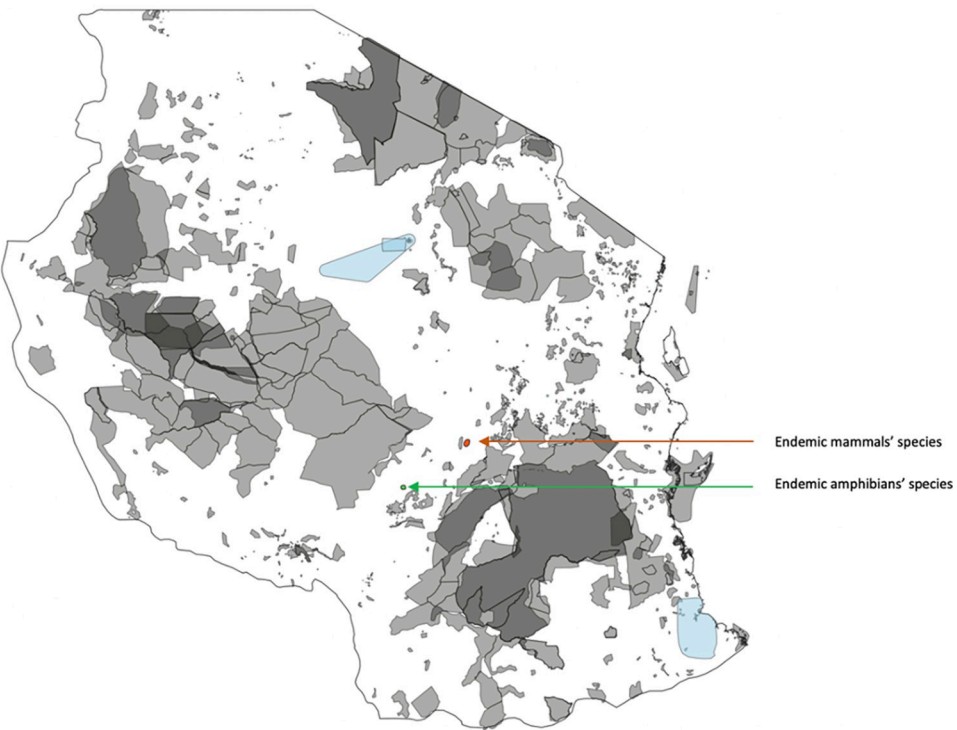

**Fig 5. Coverage of the ranges of gap and poorly covered endemic Tanzanian vertebrate species outside all the types of protected areas in the country.** Orange shows endemic mammal species, blue reptiles, and green represents endemic amphibian species. There are no gap species of birds. Protected areas are indicated as grey polygons.

ecoregions, and picking the 'best' remaining sites owned by the current management body–Tanzania Forest Service. We have shown that the first designated NFR sites, in the Eastern Arc Mountains or Coastal Forests ecoregions, have higher biodiversity values in terms of endemic and threatened species than more recently declared sites. The first declared NFRs were known to have globally important biodiversity values [10, 27, 28] and had received more research funding and survey efforts [10, 29]. Many of the newer NFRs have received little or no biodiversity survey attention, so we expect their known values to increase (see e.g., Minziro NFR, [30]). However, the more recent sites also add value and together the NFRs cover the majority of Tanzanian endemic and threatened species (Fig 2B and 2C). However, the current NFR coverage might be moving towards saturation even if our analysis also showed that there are more species endemic to Tanzania that could be capture in the enhanced network.

**Table 1. Comparison of the proportion of the range of gap and poorly covered endemic amphibian, reptile, bird, and mammal species, when all protected areas are included compared to only the NFRs.**

|  | All types of protected areas | | All NFRs | |
| --- | --- | --- | --- | --- |
|  | % gap species | % poorly covered species | % gap species | % poorly covered species |
| Amphibian species | 6.52 | 0 | 10.77 | 13.85 |
| Reptile species | 6.52 | 6.52 | 7.94 | 22.22 |
| Bird species | 0 | 0 | 0 | 30.76 |
| Mammal species | 2.17 | 0 | 5.26 | 10.53 |

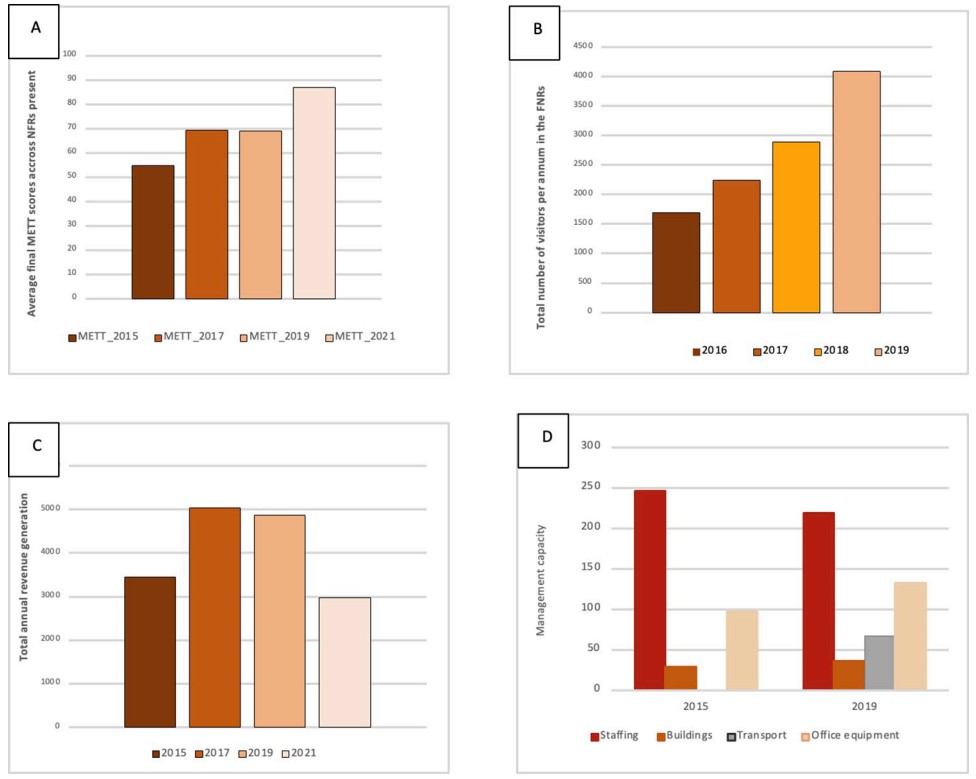

**Fig 6.** Changes in management scores from 2015–2021 for A) mean METT score across the sites; B) the annual number of tourists in all sites; C) average total revenue generated in the reserves (US dollars); D) sum of the management capacity in 2015 and 2019 comprising staffing (staff and rangers), buildings (offices and ranger posts), transport (vehicles), and office equipment.

## What gaps remain in the NFR network?

The NFR network in Tanzania compliments its world-renowned national parks system that mainly protects non-forest or drier forest habitats, with the exceptions of Udzungwa Mountains, Mahale, and Kilimanjaro. As more NFRs have been created, the proportion of Tanzanian endemic gap species has decreased for all species groups (Fig 3). This is especially true for endemic mammal and bird species, which respectively have 5% and 0% of gap species remaining. The size of the NFRs compared with the large range of some Tanzanian endemic species means that not all species are adequately covered. However, if we also include other protected areas in Tanzania then a greater proportion of ranges are included within protected lands (Fig 5).

The largest number of NFR gap species were from coastal forests (Figs 4 and 5). Even after adding all the types of protected areas in the country, the protection gap in the southeastern region of the country remains. There are a few, relatively small, forest reserves present in that region. There may be an opportunity to expand reserves, such as Rondo NFR or Matapwa Forest Reserve, to cover a part of the ranges of current gap species.

## Assessment of the management of the Nature Forest Reserves

Development of the NFRs has been funded by a combination of the Tanzanian government (through the Tanzania Forest Service) and donor funding including the Global Environment Facility, the European Union, the Eastern Arc Mountains Conservation Endowment Fund,

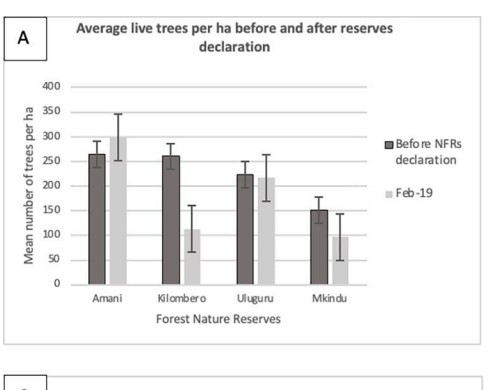
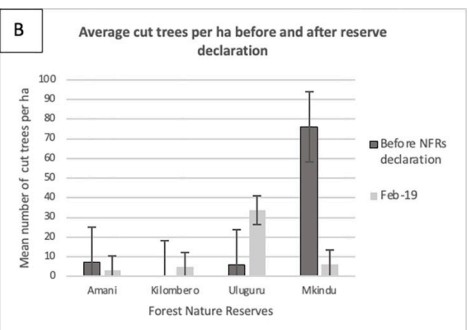
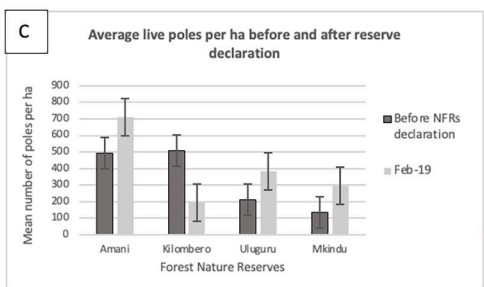
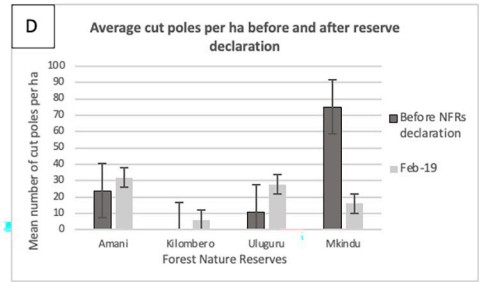

**Fig 7.** Changes in forest disturbance over time A) average living trees (>20 cm dbh) per ha, B) average cut trees per ha, C) average living poles per ha (<20 cm dbh), and D) average cut poles per ha in Amani, Kilombero, Uluguru, and Mkingu Nature Forest Reserves before reserve declaration and in February 2019.

and private donors such as the Aage V. Jensen Foundation, as well as local and international NGOs. This support has been critical to creating the current system of NFRs and has led to the observed improvement in management effectiveness (METT scores) from 2015 to 2021 (Fig 6A).

Sustainable funding of the NFR network is critical to long term management and retention of biodiversity values. The number of tourists visiting the sites increased from 2015 to 2019, but their contribution to management costs remained small (Fig 6B and 6C). Reserves such as Chome, Magamba, and Amani, with better infrastructure, accessibility, and proximity to the northern tourism circuit, had the largest growth in tourism numbers. International tourism was heavily impacted by the COVID-19 pandemic. Tourism numbers in the July-June financial year straddling 2022–2023 were greatly improved with over 240,000 tourists visiting the sites and bringing in over 550,000 USD. There remains potential for further improving local and international tourists visiting the NFRs and increasing revenue collection, but the management costs far exceed the income generation so far and care must be taken not to develop facilities that remain underused.

Forest disturbance (using the proxy of cut trees and poles) generally declined from 2001 to 2019 in three of the four studied NFRs, suggesting improved management at these sites. However, this improvement could be due to fewer suitable trees and poles to harvest: indeed, our analyses showed that the live trees density are globally decreasing in three sites of the four sites studied. In addition, evidence of harvesting could be demonstrated with an increase in the density of live poles, as these are coming back in areas where the trees have been cut. The increase in the live poles density was observed in our study in three of the four sites studied. However, this assessment is only available for four reserves, and we have no overview of how this condition may have changed in other sites. Anecdotal evidence suggests that some other reserves are experiencing considerable disturbance, including hunting pressure, that is not

captured in the available data. This is the case in Uzungwa Scarp NFR, where decades of poor management of the former Forest Reserve have led to a dramatic loss of wildlife [31, 32], and snares set to catch mammals and birds remain prevalent in 2022 (field researchers pers com). However, since its upgrading to NFR status in 2016, increased management efforts supported by local and international agencies and implemented in close partnership with TFS have led to a considerable reduction in some other threats to this reserve, like tree cutting and farmland encroachment [33]. Hunting pressure and long-term forest encroachment may have extirpated the Udzungwa forest partridge species, *Xenoperdix udzungwensis*, from the Nyumbanitu forest within the Kilombero NFR, where it was still at least until 2016 [34]. On ground insights from all other NFR would likely also contain important stories of success and remaining challenges for NFR management.

## Recommendations

Our study has shown that for a more effective coverage and protection of endemic gap species, the TFS should urgently consider further expanding the current network. This can be done both by creating new NFRs and re-gazetting existing sites to higher-level categories of conservation management. This primarily relates to south-eastern Tanzania.

To improve the management effectiveness of NFRs, we suggest that future management plans should emphasise enforcing hunting and logging regulations at all sites, raising awareness of conservation issues in local villages, and providing local people with viable alternatives to activities that impact the forests and their biodiversity.

More biodiversity monitoring, including participatory biodiversity monitoring, should be carried out to compare against management effectiveness scores. This is crucial for current and future sites for the NFRs to be effective in their purposes. Baseline biodiversity surveys are also required for sites with no or little biodiversity data (e.g., Hassama, Itulu Hill, Mwambesi, Pindiro, and Uzigua NFRs). Additional surveys of species/taxa distribution patterns are needed, and future research is needed to detect range shifts due to global warming [35] and to assess how to better prepare and manage future climate risks. Furthermore, life history information, species abundance and human uses should be added to future analyses [36].

For the main management authorities, a realistic, inflation-linked budget is necessary to afford new equipment, facilities and infrastructure, and to maintain patrols in the NFR network. To increase the budget for reserve management, we suggest the following potential revenue streams: capturing revenue from ecosystem services (such as water catchment and carbon stocks [37–39]); soliciting additional investment or donations from philanthropic donors, and increasing the revenue generated from touristic activities [10].

## Supporting information

**S1 Fig. Painting 1: Fischer's turaco (*Tauraco fischeri*) and white-starred robin (*Pogonocichla stellata*) found in the sub-montane forests of the Nature Reserves in the East Usambara and Uluguru Mountains.** Painting 2: The narrowly endemic Udzungwa forest partridge (*Xenoperdix udzungwensis*), and grey-faced sengi (*Rhynchocyon udzungwensis*) encountering each other on the forest floor in the Kilombero Nature Reserve in the Udzungwa Mountains within the Eastern Arc mountains region.
(ZIP)

**S1 Table. Online data sources used in this study: Biodiversity (species lists for plants, birds, mammals, reptiles, and amphibians per site); management (revenue generation, tourist numbers, management capacity, and forest disturbance); geospatial data (species**

**range maps, protected areas, etc.) used for the spatial analysis.**
(TIF)

**S2 Table. Basic attributes of 21 declared and one proposed Tanzanian Nature Forest Reserves.**
(ZIP)

## Acknowledgments

We thank the many staff of the Tanzania Forest Service and key donors, including the Global Environment Facility (via UNDP and the World Bank), the Critical Ecosystem Partnership Fund, the European Union, Governments of Norway, Finland and Denmark, and facilitation by several NGOs: WWF, Tanzania Forest Conservation Group, CARE International, IUCN, WCS, BirdLife International, and African Wildlife Foundation. We also want to thank Jon Fjeldså for his two paintings on some of the species occurring in the Nature Forest Reserves in Tanzania.

**Dedication:** To those who put their heart and soul into creating a network of Tanzanian Nature Reserves and died too early along the way: Alan Rodgers, John Mejissa, Corodius Sawe, Gerald Kamwenda, Peter Sumbi.

## Author Contributions

**Conceptualization:** Neil D. Burgess, Lars Dinesen, Peter Sumbi, Isaac Malugu, Roy E. Gereau, Marcelo Gonçalves de Lima.

**Data curation:** Neil D. Burgess, Lars Dinesen, Peter Sumbi, Isaac Malugu, Roy E. Gereau, Marcelo Gonçalves de Lima, Amina Akida, Evarist Nashanda, Zainabu Shabani, Yusuph Tango, Someni Mteleka, Dos Santos Silayo, Juma Mwangi, Gertrude Lyatuu, Philip J. Platts, Francesco Rovero.

**Formal analysis:** Claire Ract, Lars Dinesen, Peter Sumbi, Isaac Malugu, Roy E. Gereau, Marcelo Gonçalves de Lima, Philip J. Platts, Francesco Rovero.

**Investigation:** Neil D. Burgess, Peter Sumbi, Isaac Malugu, Julia Latham, Lucy Anderson, Roy E. Gereau, Marcelo Gonçalves de Lima, Zainabu Shabani, Yusuph Tango, Someni Mteleka, Dos Santos Silayo, Juma Mwangi, Philip J. Platts.

**Methodology:** Claire Ract, Neil D. Burgess, Lars Dinesen, Peter Sumbi, Isaac Malugu, Julia Latham, Lucy Anderson, Roy E. Gereau, Marcelo Gonçalves de Lima, Francesco Rovero.

**Project administration:** Neil D. Burgess.

**Resources:** Neil D. Burgess, Lars Dinesen, Isaac Malugu, Julia Latham, Roy E. Gereau, Zainabu Shabani, Yusuph Tango, Someni Mteleka, Dos Santos Silayo, Juma Mwangi, Philip J. Platts.

**Software:** Claire Ract.

**Supervision:** Neil D. Burgess, Roy E. Gereau.

**Validation:** Neil D. Burgess, Lars Dinesen, Julia Latham, Roy E. Gereau, Marcelo Gonçalves de Lima, Zainabu Shabani, Yusuph Tango, Someni Mteleka, Dos Santos Silayo, Juma Mwangi, Philip J. Platts, Francesco Rovero.

**Visualization:** Claire Ract, Neil D. Burgess, Philip J. Platts.

**Writing – original draft:** Claire Ract, Neil D. Burgess, Lars Dinesen, Peter Sumbi, Isaac Malugu, Julia Latham, Lucy Anderson, Roy E. Gereau, Marcelo Gonçalves de Lima, Amina Akida, Evarist Nashanda, Gertrude Lyatuu, Philip J. Platts, Francesco Rovero.

**Writing – review & editing:** Claire Ract, Neil D. Burgess, Lars Dinesen, Isaac Malugu, Julia Latham, Roy E. Gereau, Marcelo Gonçalves de Lima, Philip J. Platts, Francesco Rovero.

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
