## [Decision Letter · Decision Letter 0]

20 Mar 2023

PONE-D-23-01905Nature Forest Reserves in Tanzania and their importance for conservationPLOS ONE

Dear Dr. Ract,

Thank you for submitting your manuscript to PLOS ONE. After careful consideration, we feel that it has merit but does not fully meet PLOS ONE’s publication criteria as it currently stands. Therefore, we invite you to submit a revised version of the manuscript that addresses the points raised during the review process. Two reviewers have provided helpful comments below as well as in a marked up copy for you to download. One of the primary criteria for publication in PLOS is analytical rigor. Reviewer #1 has provided feedback on your analytical approach. You will also find suggestions for an overhaul of the framework for your analysis. I encourage you to consider the suggestion to adopt a systematic conservation planning approach. I agree that this could yield a higher impact for your work. However, I leave this to the discretion of the authors, so long as the methods employed are statistically valid and the approach clear and repeatable.

We look forward to receiving your revised manuscript.

Kind regards,

Stephanie S. Romanach, Ph.D.

Academic Editor

PLOS ONE

“The author(s) received no specific funding for this work.

We did build the paper off material collected by a GEF funded project called:

“Enhancing the Forest Nature Reserves Network for Biodiversity Conservation in Tanzania".

This is mentioned in our manuscript as well as others donors.

Funders had no role in the study design, data collection, analysis, conclusion, decision to publish the manuscript as it stands.”

“The authors have declared that no competing interests exist.

The work is all original research carried out by the authors. All authors agree with the contents of the manuscript and its submission to the journal.

No part of the research has been published in any form elsewhere, unless it is fully acknowledged in the manuscript.

This manuscript is not being considered for publication elsewhere while it is being considered for publication in this journal.

Any research in the paper not carried out by the authors is fully acknowledged in the manuscript. Neil Burgess worked on the Global Environment Facility (GEF) project “Enhancing the Forest Nature Reserves Network for Biodiversity Conservation in Tanzania”, that collected some of the information. However, he did not collect the data.

All sources of funding are acknowledged in the manuscript, and the authors have declared any direct financial benefits that could result from publication.”

4. We note that Figures 1,6 and 7 in your submission contain [map/satellite] images which may be copyrighted. All PLOS content is published under the Creative Commons Attribution License (CC BY 4.0), which means that the manuscript, images, and Supporting Information files will be freely available online, and any third party is permitted to access, download, copy, distribute, and use these materials in any way, even commercially, with proper attribution. For these reasons, we cannot publish previously copyrighted maps or satellite images created using proprietary data, such as Google software (Google Maps, Street View, and Earth). For more information, see our copyright guidelines: http://journals.plos.org/plosone/s/licenses-and-copyright.

a. You may seek permission from the original copyright holder of Figures 1,6 and 7 to publish the content specifically under the CC BY 4.0 license. 

5. Please upload a copy of Supporting Information Table S1 which you refer to in your text on page 9.

Reviewers' comments:

Reviewer's Responses to Questions

**Comments to the Author**

1. Is the manuscript technically sound, and do the data support the conclusions?

Reviewer #1: Partly

Reviewer #2: Yes

2. Has the statistical analysis been performed appropriately and rigorously? 

Reviewer #1: No

Reviewer #2: Yes

3. Have the authors made all data underlying the findings in their manuscript fully available?

Reviewer #1: No

Reviewer #2: Yes

4. Is the manuscript presented in an intelligible fashion and written in standard English?

Reviewer #1: Yes

Reviewer #2: Yes

5. Review Comments to the Author

Reviewer #1: This is a useful and encouraging review of the conservation status of forest reserves and their vertebrate wildlife in Tanzania. The manuscript reads like a report rather than a critical review and conclusions are framed as suggestions. A more decisive statement of priorities to ensure positive outcomes for the NFR network is needed. The statistical analyses rely on non-parametric correlation, which is incorrect because time is incorrectly used as the independent variable when, in fact, it is a factor (i.e., the year in which a METT assessment was done). ANOVA or its non-parametric equivalents (Kruskal-Wallis; Friedmann’s test) are more appropriate but see below for comments about using a systematic conservation planning approach. Stacked bar-graphs are difficult to interpret in some cases and another graphic method may be more useful. Maps lack legends and scales. The writing is reasonable but can be finessed and improved. I have done an extensive edit of the manuscript using tracked changes and attach it too my review.

That said, I think this is a useful paper and could be published but falls way short of its potential. I would have liked to see more critical assessment and encourage the authors to consider upgrading their analyses. This can be achieved by using well established iterative algorithms that allow for systematic identification of priority reserves based on species rarity and complementarity. The latter would up-date the research to contemporary practices. See the following references:

Margules, C. R., & Pressey, R. L. (2000). Systematic conservation planning. Nature, 405, 243-253.

McIntosh, E. J., Pressey, R. L., Lloyd, S., Smith, R. J., & Grenyer, R. (2017). The Impact of Systematic Conservation Planning. Annual Review of Environment and Resources, 42(1), 677-697. doi:10.1146/annurev-environ-102016-060902

Kukkala, A. S., & Moilanen, A. (2013). Core concepts of spatial prioritisation in systematic conservation planning. Biological Reviews, 88(2), 443-464. doi: 10.1111/brv.12008

Watson, J. E., Grantham, H. S., Wilson, K. A., & Possingham, H. P. (2011). Systematic conservation planning: past, present and future. In R. J. Ladle & R. J. Whittaker (Eds.), Conservation biogeography (pp. 136-160): Blackwell Publishing Ltd.

Reside, A. E., Butt, N., & Adams, V. M. (2018). Adapting systematic conservation planning for climate change. Biodiversity and Conservation, 27(1), 1-29. doi:10.1007/s10531-017-1442-5

Eeley, H. A. C., Lawes, M. J., & Reyers, B. (2001). Priority areas for the conservation of subtropical indigenous forest in southern Africa: a case study from KwaZulu-Natal. Biodiversity and Conservation, 10, 1221-1246.

Without analyses that provide insights to which reserves are a priority for conservation, the conservation conclusions are platitudinous and not implementable. The suggestion that “…we recommend that TFS consider further expanding the current network to include the small number of endemic gap and poorly covered species that are outside the current NFR network, but could be included with some management status changes at existing sites.” is the sort of recommendation I was expecting and could be elicited by a systematic conservation planning approach.

Good luck.

Mike Lawes

Reviewer #2: 1. Summary of the research and your overall impression

Tanzania has expanded its network of Nature Forest Reserves to include almost all forest types. The study found that, the change of management has increased conservation effectiveness especially where donor funds are available. However, there are still some management challenges include illegal logging, charcoal production, firewood, pole cutting, hunting and wildfires. The study recommends areas for further research and strategies to further improve the management effectiveness.

• Some of the presented information need to be updated (I will provide examples in the below Section).

• The recommendations provided to improve management effectiveness are pertinent but need to consider current issues such as putting the NFRs into carbon credit trading schemes and the use of novel high technologies such as drones in managing the NFRs.

2. Evidence and examples

In order to improve the manuscript the authors should:

Line 73: TFS started its operations in 2011

Line 77: NFRs in Tanzania currently contains 22 reserves (the three that were under proposed status and Vikindu NFR that is missing in the list and the Map). Please update this information in the entire document.

6. PLOS authors have the option to publish the peer review history of their article (what does this mean?). If published, this will include your full peer review and any attached files.

Reviewer #1: **Yes: **Michael J Lawes

Reviewer #2: No

---

## [Author Response · Author response to Decision Letter 0]

3 Oct 2023

Editor comment concerning the financial disclosure. Response: as no authors have received funding for this study we stated in the cover letter: “The authors received no specific funding for this work.”

Editor comment concerning the competing Interests section: as we did not have any competing interests we stated: "The authors have declared that no competing interests exist." in the cover letter. 

Editor comment: "We note that Figures 1,6 and 7 in your submission contain [map/satellite] images which may be copyrighted. All PLOS content is published under the Creative Commons Attribution License (CC BY 4.0), which means that the manuscript, images, and Supporting Information files will be freely available online, and any third party is permitted to access, download, copy, distribute, and use these materials in any way, even commercially, with proper attribution. For these reasons, we cannot publish previously copyrighted maps or satellite images created using proprietary data, such as Google software (Google Maps, Street View, and Earth). For more information, see our copyright guidelines: http://journals.plos.org/plosone/s/licenses-and-copyright". Response: "We filled out 3 content permission forms for figures 1, 6 and 7 (now Figure 1, 4, and 5 as we changed the order of the figures in the revised manuscript). 

Editor comment: "Please upload a copy of Supporting Information Table S1 which you refer to in your text on page 9". Response: "We removed table S1 from the manuscript, as we made the full data available. However, we uploaded all the tables in the manuscript as supporting information". 

Reviewer 1 comments: "This is a useful and encouraging review of the conservation status of forest reserves and their vertebrate wildlife in Tanzania. The manuscript reads like a report rather than a critical review and conclusions are framed as suggestions. A more decisive statement of priorities to ensure positive outcomes for the NFR network is needed. The statistical analyses rely on non-parametric correlation, which is incorrect because time is incorrectly used as the independent variable when, in fact, it is a factor (i.e., the year in which a METT assessment was done). ANOVA or its non-parametric equivalents (Kruskal-Wallis; Friedmann’s test) are more appropriate but see below for comments about using a systematic conservation planning approach. Stacked bar-graphs are difficult to interpret in some cases and another graphic method may be more useful. Maps lack legends and scales. The writing is reasonable but can be finessed and improved. I have done an extensive edit of the manuscript using tracked changes and attach it too my review.

That said, I think this is a useful paper and could be published but falls way short of its potential. I would have liked to see more critical assessment and encourage the authors to consider upgrading their analyses. This can be achieved by using well established iterative algorithms that allow for systematic identification of priority reserves based on species rarity and complementarity. The latter would up-date the research to contemporary practices. See the following references:

Margules, C. R., & Pressey, R. L. (2000). Systematic conservation planning. Nature, 405, 243-253.

McIntosh, E. J., Pressey, R. L., Lloyd, S., Smith, R. J., & Grenyer, R. (2017). The Impact of Systematic Conservation Planning. Annual Review of Environment and Resources, 42(1), 677-697. doi:10.1146/annurev-environ-102016-060902

Kukkala, A. S., & Moilanen, A. (2013). Core concepts of spatial prioritisation in systematic conservation planning. Biological Reviews, 88(2), 443-464. doi: 10.1111/brv.12008

Watson, J. E., Grantham, H. S., Wilson, K. A., & Possingham, H. P. (2011). Systematic conservation planning: past, present and future. In R. J. Ladle & R. J. Whittaker (Eds.), Conservation biogeography (pp. 136-160): Blackwell Publishing Ltd.

Reside, A. E., Butt, N., & Adams, V. M. (2018). Adapting systematic conservation planning for climate change. Biodiversity and Conservation, 27(1), 1-29. doi:10.1007/s10531-017-1442-5

Eeley, H. A. C., Lawes, M. J., & Reyers, B. (2001). Priority areas for the conservation of subtropical indigenous forest in southern Africa: a case study from KwaZulu-Natal. Biodiversity and Conservation, 10, 1221-1246.

Without analyses that provide insights to which reserves are a priority for conservation, the conservation conclusions are platitudinous and not implementable. The suggestion that “…we recommend that TFS consider further expanding the current network to include the small number of endemic gap and poorly covered species that are outside the current NFR network, but could be included with some management status changes at existing sites.” is the sort of recommendation I was expecting and could be elicited by a systematic conservation planning approach.

Good luck.

Mike Lawes". 

Response: "We have re-drafted the text of the paper and tried to sharpen it as a scientific paper. The data has been made fully available. We also have re-made all the statistics in the paper using suggested approaches (we used the Friedman test as suggested). Concerning the stacked bar graphs, we have reviewed the figures and where possible we have changed the format, and we have reviewed the legends and labeling and tried to improve this throughout the paper. Concerning the writing, we have accepted the proposed edits with thanks and sought to further improve the language throughout the paper. 

Concerning our analyses: we are aware of the systematic conservation planning literature and several of the authors have used these approaches at various geographical scales. We have published papers using these approaches over the past 25 years. We believe our analytical approach is sound because species endemic to Tanzania with small ranges would drive complementarity analyses and have sought to greatly clarify what we have done in light of this comment. 

We have clarified in the paper that we have done 2 main analyses of conservation priority and we have re-run the analyses using the latest data on the number of Nature Forest Reserves and other reserves in Tanzania. 

1) The first analysis uses the species lists available from the Nature Forest Reserves themselves. This is a new database, and the plant database took over 3 years to create by an expert. We have used these data to rank the values of the reserves. 

2) The second analysis uses spatial data for all vertebrates in Tanzania, with an emphasis on endemic and threatened species. For these data we apply a sequential approach to determining the species already present within the existing Nature Forest Reserves, and then within other kinds of reserves in the country. This approach provides a clear set of priorities to TFS on the remaining species that occur outside the protected areas. We feel this is more operationally useful than a systematic conservation planning approach that does not necessarily consider the existing reserves and their sequence of creation and success.

Therefore, we have sought to clarify our approach. We use distribution data for all Tanzania endemic species in our analysis and have undertaken a stepwise analysis where we excluded from the analysis all species covered by the Nature Forest Reserves, and then all other kinds of reserves, to show the remaining gaps. As our paper was focused on the existing network of Nature Reserves and their development over time, we sought to provide an assessment of the ‘success’ of this network and provide guidance to TFS who are seeking to further expand the network to critical areas. We have clarified the approach taken. And we have provided clearer guidance to TFS. 

Through the updating process, a series of TFS (Tanzania government) authors have asked to be included in the paper, which we think is testimony to the success of our approach in terms of delivering an impact on reserve creation and management in Tanzania". 

Reviewer 2 comments: " Summary of the research and your overall impression

Tanzania has expanded its network of Nature Forest Reserves to include almost all forest types. The study found that, the change of management has increased conservation effectiveness especially where donor funds are available. However, there are still some management challenges include illegal logging, charcoal production, firewood, pole cutting, hunting and wildfires. The study recommends areas for further research and strategies to further improve the management effectiveness.

• Some of the presented information need to be updated (I will provide examples in the below Section).

• The recommendations provided to improve management effectiveness are pertinent but need to consider current issues such as putting the NFRs into carbon credit trading schemes and the use of novel high technologies such as drones in managing the NFRs.

2. Evidence and examples

In order to improve the manuscript the authors should:

Line 73: TFS started its operations in 2011

Line 77: NFRs in Tanzania currently contains 22 reserves (the three that were under proposed status and Vikindu NFR that is missing in the list and the Map). Please update this information in the entire document". 

Response: "We added several references on REDD+: We added papers on: (i) a document REDD + progress and challenges in Tanzania (ii) a document published by the Tanzania government and his project to make Tanzania REDD+ ready for the implementation of Paris Agreement and to contributes to the country reduction of carbon emissions (iii) a local NGO in the country called “carbon Tanzania”, developing forest conservation projects in collaboration with forest communities to generate forest carbon credits and allows forest communities to earn revenues in return for measurable and verifiable forest protection activities: citations 37, 38 and 39. 

Line 73: we changed TFS starting operations year to 2011. 

Line 77: We updated the number of reserves: we worked closely with the government of Tanzania, and we learned that the network now has 22 reserves declared and 4 proposed: Nou, East Matogoro, Kindoroko, and Rau. Vikindu is called by another name: Pugh Kazimzumwi, a reserve that we already have in the list and the map. All the other information was updated in the entire document”.

---

## [Editor Report · Decision Letter 1]

11 Oct 2023

Nature Forest Reserves in Tanzania and their importance for conservation

PONE-D-23-01905R1

Dear Dr. Ract,

We’re pleased to inform you that your manuscript has been judged scientifically suitable for publication and will be formally accepted for publication once it meets all outstanding technical requirements.

Kind regards,

Stephanie S. Romanach, Ph.D.

Academic Editor

PLOS ONE

---

## [Editor Report · Acceptance letter]

28 Jan 2024

PONE-D-23-01905R1 

PLOS ONE

Dear Dr. Ract, 

I'm pleased to inform you that your manuscript has been deemed suitable for publication in PLOS ONE. Congratulations! Your manuscript is now being handed over to our production team.

Kind regards, 

on behalf of

Dr. Stephanie S. Romanach 

Academic Editor

PLOS ONE